# Design and Evaluation of Chimeric *Plasmodium falciparum* Circumsporozoite Protein-Based Malaria Vaccines

**DOI:** 10.3390/vaccines12040351

**Published:** 2024-03-25

**Authors:** William H. Stump, Hayley J. Klingenberg, Amy C. Ott, Donna M. Gonzales, James M. Burns

**Affiliations:** Center for Molecular Parasitology, Department of Microbiology and Immunology, Drexel University College of Medicine, 2900 Queen Lane, Philadelphia, PA 19129, USA; ws454@dragons.drexel.edu (W.H.S.); hayley.klingenberg@ucsf.edu (H.J.K.); ac3482@drexel.edu (A.C.O.); donna.gonzales@pennmedicine.upenn.edu (D.M.G.)

**Keywords:** malaria vaccine, circumsporozoite protein, *Plasmodium falciparum*, chimeric

## Abstract

Efficacy data on two malaria vaccines, RTS,S and R21, targeting *Plasmodium falciparum* circumsporozoite protein (*Pf*CSP), are encouraging. Efficacy may be improved by induction of additional antibodies to neutralizing epitopes outside of the central immunodominant repeat domain of *Pf*CSP. We designed four r*Pf*CSP-based vaccines in an effort to improve the diversity of the antibody response. We also evaluated *P. falciparum* merozoite surface protein 8 (*Pf*MSP8) as a malaria-specific carrier protein as an alternative to hepatitis B surface antigen. We measured the magnitude, specificity, subclass, avidity, durability, and efficacy of vaccine-induced antibodies in outbred CD1 mice. In comparison to N-terminal- or C-terminal-focused constructs, immunization with near full-length vaccines, r*Pf*CSP (#1) or the chimeric r*Pf*CSP/8 (#2), markedly increased the breadth of B cell epitopes recognized covering the N-terminal domain, junctional region, and central repeat. Both r*Pf*CSP (#1) and r*Pf*CSP/8 (#2) also elicited a high proportion of antibodies to conformation-dependent epitopes in the C-terminus of *Pf*CSP. Fusion of *Pf*CSP to *Pf*MSP8 shifted the specificity of the T cell response away from *Pf*CSP toward *Pf*MSP8 epitopes. Challenge studies with transgenic *Plasmodium yoelii* sporozoites expressing *Pf*CSP demonstrated high and consistent sterile protection following r*Pf*CSP/8 (#2) immunization. Of note, antibodies to conformational C-terminal epitopes were not required for protection. These results indicate that inclusion of the N-terminal domain of *Pf*CSP can drive responses to protective, repeat, and non-repeat B cell epitopes and that *Pf*MSP8 is an effective carrier for induction of high-titer, durable anti-*Pf*CSP antibodies.

## 1. Introduction

There have been measurable gains in reducing the global malaria burden through the integration of control programs involving insecticide-treated bed nets, indoor spraying of residual insecticides, rapid diagnosis, treatment with multi-drug combinations, and intermittent preventative therapy [1]. Nevertheless, the most recent World Malaria Report indicates that the preceding decline in clinical cases of malaria and malaria-related deaths is leveling off in most regions and is even showing signs of reversal in others [2]. An efficacious malaria vaccine would be a valuable tool to add to control efforts, with a multivalent, multistage formulation being most desirable [3,4,5]. In addition to the complexity of immune-mediated protection against *Plasmodium falciparum*, challenges for the subunit malaria vaccine effort relate to scalable production of high-quality recombinant vaccine candidates, polymorphism of T and B cell epitopes, immunogenicity and durability of vaccine-induced immune responses, antigenic competition, and adjuvant selection [6,7]. Many of these were encountered during the development of RTS,S, the most advanced *P. falciparum* subunit malaria vaccine, which partially protected children in a Phase III clinical trial [8,9,10,11,12,13]. However, protection elicited by this pre-erythrocytic-stage vaccine waned and the lack of vaccine- and/or infection-induced immunity to blood-stage parasites left these children susceptible to clinical disease. It is encouraging that increases in efficacy and durability of protection have been achieved with the more recently developed R21 vaccine, a modified version of RTS,S, formulated with Matrix-M as adjuvant [14,15,16]. With a positive recommendation by the World Health Organization for vaccination of children with RTS,S and R21, the deployment of these vaccines in several African countries was initiated and is expanding [17,18].

RTS,S and R21 contain a portion of the immunodominant central repeat domain of the *P. falciparum* circumsporozoite protein (*Pf*CSP) along with the C-terminus containing T cell epitopes and an adhesive thrombospondin type-I repeat (TSR) [19,20,21]. Recent studies mapped additional neutralizing B cell epitopes to the junction of the N-terminal domain and the central repeat domain [22,23,24,25]. Earlier work also demonstrated that antibodies which block proteolytic cleavage of *Pf*CSP at a site in the N-terminal domain prevent infection [26]. These protective B cell epitopes are lacking in RTS,S and R21. Immunization with *Pf*CSP vaccine constructs that include the N-terminus, a portion of the central repeat, and the C-terminus of *Pf*CSP may drive IgG responses to a broader set of conserved B cell epitopes and increase efficacy. At the same time, earlier data demonstrating that the N-terminus of *Pf*CSP can fold to mask the adhesive domain of the C-terminus and facilitate the exit of sporozoites out of the dermis [27,28] must also be considered.

Historically, carrier proteins have been of value in overcoming some challenges related to vaccine immunogenicity [29,30]. Most commonly, these are heterologous protein carriers unrelated to the pathogen of interest. This is the case with the use of Hepatitis B surface Antigen as a carrier, fused to a truncated *Pf*CSP molecule in the RTS,S vaccine and more recently in the related R21 construct [19,20,21]. Building on extensive in vivo immunogenicity and efficacy data with monovalent *Plasmodium yoelii* merozoite surface protein 1_42_ (MSP1_42_) and merozoite surface protein 8 (MSP8) vaccines and a chimeric *P. yoelii* MSP1_19_ + MSP8 vaccine [31,32], we developed the highly conserved, highly immunogenic *Pf*MSP8 as a carrier for malaria vaccine delivery [33,34]. By genetically fusing targeted neutralizing B cell epitopes to *Pf*MSP8, we overcame challenges associated with the production of recombinant antigen (rAg) vaccines (conformation, quality, yield) and suboptimal immunogenicity. We demonstrated its utility as a carrier for blood-stage vaccine candidates including *Pf*MSP1 and *Pf*MSP2, and for the structurally complex 25 kDa transmission blocking vaccine candidate, *Pf*s25. For *Pf*MSP1_19_, fusion to *Pf*MSP8 facilitated production of a vaccine in high yield and appropriate conformation that elicited strong CD4+ T cell help for the production of merozoite neutralizing antibodies to linked *Pf*MSP1_19_ epitopes in outbred strains of mice, rabbits, and non-human primates [34,35]. Elicited antibodies were highly cross-reactive between FVO and 3D7 alleles of *Pf*MSP1_19_ and potently inhibited the in vitro growth of *P. falciparum* blood-stage parasites. For *Pf*MSP2 vaccines, fusion to *Pf*MSP8 prevented formation of *Pf*MSP2 fibrils and masking of B cell epitopes in *Pf*MSP2, which enhanced production of *Pf*MSP2-specific antibodies in mice and rabbits that opsonized merozoites for uptake by macrophages [36,37]. Immunization with chimeric *Pf*MSP2/8 formulated with glucopyranosyl lipid adjuvant-stable emulsion (GLA-SE, a synthetic TLR4 agonist) as adjuvant elicited high *Pf*MSP2-specific antibody titers that persisted for at least 12 months. As a carrier, *Pf*MSP8 facilitated production of a *Pf*s25-based vaccine in high yield and appropriate conformation using simplified purification protocols and enhanced the immunogenicity of *Pf*s25 when adjuvanted with GLA-SE [38,39]. *Pf*s25/8-induced IgG (rabbit, mouse) was functional, potently reducing transmission of *P. falciparum* parasites to mosquitoes when measured using the standard membrane feeding assay (SMFA). Critical for the development of multivalent, multistage malaria vaccines, fusion to *Pf*MSP8 was required to maintain the immunogenicity of *Pf*MSP1_19_, *Pf*MSP2, and *Pf*s25 when formulated in combination [37].

Here, we build on knowledge gained from our own vaccine studies and from the wealth of information revealed by studies (old and new) of protective immune responses to *Pf*CSP. We tested the hypotheses that (i) immunization with r*Pf*CSP-based vaccines containing N-terminal, central repeat, and C-terminal domains can elicit responses to a more diverse set of B cell epitopes (repeat and non-repeat domains) and (ii) *Pf*MSP8 can function as an effective carrier protein for r*Pf*CSP to drive protective and durable antibody responses.

## 2. Materials and Methods

### 2.1. Design, Expression, and Purification of rPfCSP-Based Vaccines

Four r*Pf*CSP vaccine antigens were produced based on the *Pf*CSP sequence of the 7G8 strain of *P. falciparum* [19]. Construct #1 (r*Pf*CSP) contained amino acids 27-159 (N-terminal domain + first 9 repeats) plus amino acids 248-391 (10 repeats + C-terminal domain) (GenBank Accession # PP331495). Construct #2 (r*Pf*CSP/8) contained the same *Pf*CSP sequence as construct #1, fused to the *Pf*MSP8 carrier [34,39] (GenBank Accession # PP331496). Construct #3 (r*Pf*CSPN/8) included only amino acids 27-159, spanning the N-terminal domain and 9 repeats (6 NANP, 3 NVDP) fused to *Pf*MSP8 (GenBank Accession # PP331497). Construct #4 (r*Pf*CSPC/8) encoded only amino acids 248-391 (10 repeats + C-terminal domain) fused to the *Pf*MSP8 carrier (GenBank Accession # PP331498). The sequence of *Pf*CSP was codon harmonized for expression in *E. coli* [40]. Harmonized gene segments were commercially synthesized, sequenced (Blue Heron Biotechnology Inc., Bothell, WA, USA), and subcloned into our pET28-MCS-*Pf*MSP8 (CΔS) plasmid, 5′ to the *Pf*MSP8 gene [39]. For chimeric constructs, *Pf*CSP gene sequences were inserted in frame with the *Pf*MSP8 gene with a glycine–serine linker (GGSGSG) at the junction. For non-fused *Pf*CSP, two stop codons were incorporated into the 3′ end of the *Pf*CSP segment. All constructs included a 6x-His tag at the N-terminus. Expression plasmids were sequenced and transformed into SHuffle^®^ T7 Express lysY competent *E. coli* cells (New England Biolabs, Ipswich, MA, USA). This strain (i) lacks glutaredoxin reductase and thioredoxin reductase genes (Δgor ΔtrxB), allowing for disulfide bond formation, and (ii) expresses a cytoplasmic disulfide bond isomerase (DsbC) to promote proper folding [41].

Bacteria expressing each r*Pf*CSP-based vaccine antigen were grown at 30 °C in defined media in a 5.0 L culture vessel using a BioFlo115 bench-top bioreactor (New Brunswick Scientific, Edison, NJ, USA) as described [33,34]. Antigen expression was induced by the addition of isopropyl β-D-1-thiogalactopyranoside to a final concentration of 1 mM. Three hours post-induction, bacteria were harvested by centrifugation and pellets were stored at −80 °C until use. For purification of r*Pf*CSP/8 (#2), r*Pf*CSPN/8 (#3), and r*Pf*CSPC/8 (#4), bacteria pellets were lysed by resuspension in BugBuster^®^ HT protein extraction reagent in the presence of Benzonase Nuclease^®^ and recombinant Lysozyme^TM^ (EMD Millipore Corp., Burlington, MA, USA). Inclusion bodies were pelleted, washed, and solubilized as previously described for chimeric *Pf*s25/8 [39] and purified by nickel chelate affinity chromatography under non-denaturing conditions in Binding Buffer (20 mM Tris-HCl, pH 7.9, 500 mM NaCl, 5 mM imidazole) containing 5 mM reduced glutathione and 0.2% sarkosyl. For purification of r*Pf*CSP (#1), the soluble supernatant following bacterial lysis was recovered and fractionated by ammonium sulfate precipitation. r*Pf*CSP (#1) was then isolated from the 20–50% ammonium sulfate fraction by nickel-chelate affinity chromatography in Binding Buffer containing 5 mM reduced glutathione and 0.2% sarkosyl. A column wash with Binding Buffer containing 5 mM reduced glutathione and 4 M guanidine-HCl was added prior to elution. For all constructs, eluted fractions containing r*Pf*CSP vaccine antigens were pooled and dialyzed against 20 mM Tris-HCl, pH 7.9, 500 mM NaCl, 5 mM reduced glutathione and 0.2% sarkosyl. Protein concentrations were determined by bicinchoninic acid (BCA) protein assay (Pierce^TM^, ThermoFisher Scientific, Waltham, MA, USA). Protein purity and conformation were assessed by Coomassie blue staining following SDS-PAGE (reduced vs. non-reduced). Immunoblot analysis with mAbs 4B3 and 4C2 [42] (obtained from Dr. Patrick Duffy, Laboratory of Malaria Immunology and Vaccinology, NIH-NIAID) was used to confirm correct disulfide bond-dependent conformation of the C-terminal domain of r*Pf*CSP vaccines. For comparative immunogenicity studies and assessment of antibody recognition of disulfide dependent epitopes, reduced and alkylated (R/A) r*Pf*CSP (#1) and R/A r*Pf*CSP/8 (#2) were prepared by sequential treatment with 25 mM dithiothreitol overnight at 4 °C and 1 h at 37 °C followed by 125 mM iodoacetic acid for 1 h at 37 °C prior to dialysis as above. r*Pf*MSP8 was purified as reported [33].

### 2.2. Mice and Immunizations

Five- to six-week-old male and female CD1 mice were purchased form Charles River Laboratories (Wilmington, MA, USA). All animals were housed under specific pathogen-free conditions in the Animal Care Facility of Drexel University College of Medicine. Animal use protocols were reviewed, approved, and conducted in compliance with Drexel University’s Institutional Animal Care and Use Committee (Protocol #20900). Six comparative immunogenicity and efficacy studies were completed. In each study, groups of outbred CD1 mice (*n* = 10, five males, five females) were immunized s.c. with 2.5 µg/dose r*Pf*CSP-based vaccine formulated with 5 µg/dose of GLA-SE (glucopyranosyl lipid adjuvant, a synthetic TLR4 agonist, in a stable squalene-in-water emulsion, Access to Advanced Health Institute, Seattle, WA, USA) [43,44]. Control groups were immunized with GLA-SE alone. Four weeks following the primary immunization, mice received a second immunization with the same dose of antigen and adjuvant.

Immunization study #1 compared r*Pf*CSP (#1) and r*Pf*CSP/8 (#2). Immunization study #2 compared r*Pf*CSPN/8 (#3) and r*Pf*CSPC/8 (#4). In both studies, sera were collected four weeks following the boost for analysis of antibody responses. Immunization study #3 compared r*Pf*CSP (#1), r*Pf*CSP/8 (#2), r*Pf*CSPN/8 (#3), and r*Pf*CSPC/8 (#4). Splenocytes and sera were harvested four weeks following the boost for analysis of T cell and antibody responses. Immunization study #4 assessed immunogenicity and efficacy of r*Pf*CSP (#1), r*Pf*CSP/8 (#2), r*Pf*CSPN/8 (#3), and r*Pf*CSPC/8 (#4) vaccines. Immunization study #5 assessed immunogenicity and efficacy of r*Pf*CSP (#1), R/A r*Pf*CSP (#1), r*Pf*CSP/8 (#2), and R/A r*Pf*CSP/8 (#2) vaccines. In both efficacy studies, sera samples (50 µL) were collected three weeks after the boost. Mice were then infected with transgenic *P. yoelii* sporozoites expressing *P. falciparum* CSP (described below). Immunization study #6 assessed the durability of antibody responses induced by immunization with r*Pf*CSP (#1), r*Pf*CSP/8 (#2), r*Pf*CSPN/8 (#3), and r*Pf*CSPC/8 (#4). Beginning four weeks following the boost, sera samples were collected at monthly intervals for six months for analysis of antibody responses.

### 2.3. Analysis of Humoral Responses

*Magnitude:* Antigen-specific antibody titers were determined by ELISA as previously described [34,37,38]. Briefly, Costar high-binding ELISA plates (Corning Inc, Kennebunk, ME, USA) were coated with 0.25 µg/well of r*Pf*CSP (#1), reduced and alkylated r*Pf*CSP (#1), or r*Pf*MSP8 and incubated with 2-fold serial dilutions of mouse sera. Bound antibody was detected with HRP-conjugated rabbit anti-mouse IgG (H+L) (Invitrogen, Waltham, MA, USA) and ABTS (2,2′-azino-di-(3-ethylbenzothiazoline-6-sulfonate) (SeraCare Life Sciences, Inc., Milford, MA, USA) as substrate. High-titer pooled immunization sera included on each plate were used to normalize data between plates. OD_405_ values between 0.1 and 1.0 were plotted against serum dilution factor and titer was calculated as the reciprocal of the dilution that yielded an OD_405_ of 0.5. The proportion of antibodies in each serum that recognized linear epitopes was calculated as (IgG titer against R/A r*Pf*CSP (#1) divided by IgG titer against r*Pf*CSP (#1)) × 100.

*IgG Avidity:* The avidity of antigen-specific antibodies was estimated by ELISA based on the ability of antibodies to remain bound to r*Pf*CSP (#1) in the presence of increasing concentrations of ammonium thiocyanate (0.5 M to 4.0 M) as previously described [37,45]. Serum from each animal was diluted to yield an OD_405_ of ~1.0 in the absence of ammonium thiocyanate. OD_405_ was plotted against increasing concentrations of ammonium thiocyanate. The line of best fit equation was used to determine an avidity index (AI) for each serum sample based on the concentration of ammonium thiocyanate needed to reduce the amount of bound antibody by 50%.

*IgG subclass:* To assess the profile of IgG subclasses induced by immunizations, serum samples were titered on ELISA plates coated with r*Pf*CSP (#1) as previously described [31,37,38]. Bound antibodies were detected with HRP-conjugated goat antibodies specific for mouse IgG1, IgG2a, IgG2b, IgG2c, and IgG3 (Southern Biotech, Birmingham, AL, USA) with ABTS as substrate. Each plate included a standard curve of purified IgG subclass-specific mouse myeloma proteins. IgG subclass concentration was expressed as units/mL where 1 U/mL is equivalent to 1 µg/mL myeloma standard.

*Epitope specificity:* A set of 37 N-terminally biotinylated *Pf*CSP peptides (15-mers, 8 amino acid overlap) spanning the length of r*Pf*CSP (#1) was synthesized (Genscript Biotech Corp., Piscataway, NJ, USA) (Appendix A). Individual peptides (0.25 µg/well) were captured on streptavidin-coated ELISA plates (Pierce^TM^, ThermoFisher Scientific). Mouse sera were assayed at a dilution of 1:1000. Bound antibody was detected with rabbit anti-mouse IgG (H+L)-HRP and ABTS as substrate. On each plate, serial dilutions of a high-titer pool of sera from r*Pf*CSP/8 (#2)-immunized mice were assayed on peptide #22 (3 NANP core repeats) and used for normalization between plates. Immunization sera were considered reactive if the mean OD_405_ in peptide coated wells was greater than mean OD_405_ + 3 standard deviations of GLA-SE control sera. Following an initial screening assay with pooled sera from immunization studies #1 and #2 (above), 13 informative peptides spanning the N-terminal domain, junctional region, major and minor repeat region, and the C-terminus of *Pf*CSP were selected to assess the epitope specificity of antibodies in individual sera from immunized mice.

### 2.4. Analysis of Cellular Responses

Four weeks following the second immunization, spleens were harvested, and single-cell suspensions were prepared as described [31,37,38]. To assess antigen-specific cytokine production, splenocytes were stimulated with peptide pools covering the C-terminal domain of *Pf*CSP (3 pools, 6 peptides per pool, 3 µg/mL of each peptide per pool), or with r*Pf*CSP (#1) or r*Pf*MSP8 at 10 µg/mL for 96 h. Concanavalin A (1 µg/mL) and cells alone were used as positive and negative controls, respectively. Cell-free culture supernatants were collected and stored at −80 °C until analysis. IFNγ released by antigen-stimulated T cells was measured using the BD OptEIA Mouse IFNγ ELISA kit according to the manufacturer’s protocol (BD Biosciences, Inc., San Diego, CA, USA). Antigen-induced IFNγ was quantified based on a standard curve run on each assay plate.

### 2.5. Vaccine Efficacy

To test vaccine efficacy, mice were challenged with transgenic *P. yoelii* 17X sporozoites engineered to express *P. falciparum* CSP from the endogenous locus [46] (provided by Dr. Stephen Hoffman, Sanaria, Inc., Rockville, MD, USA). Cryopreserved sporozoites stored in liquid nitrogen vapor phase were quickly thawed, diluted in RPMI 1640 containing 20 mM MOPS, pH 7.4, and 1% pooled normal mouse sera. Immunized and control mice were infected by i.v. injection of 6250 sporozoites (100 µL) via the retro-orbital plexus. To determine sterile protection, infections were monitored daily for 14 days by microscopy of thin tail-blood smears stained with Giemsa. Mice were considered infected on the day in which two or more blood-stage parasites in 50 fields (1000×) were detected, with progression of infection observed on subsequent days.

### 2.6. Statistical Analysis

Statistical analysis utilized Graph Pad Prism, version 10.1.2 (GraphPad Software Inc., Boston, MA, USA). For direct comparison of cytokine and antibody responses of unrelated groups, the Mann–Whitney non-parametric test was used. For comparison of antibody responses in more than two groups, the Kruskal–Wallis test was used followed by Dunn’s test to correct for multiple comparisons. For analysis of boosting and the durability of antibody responses using paired serum samples, the Wilcoxon matched pairs signed-rank test was used. Differences in sterile protection induced by *Pf*CSP-based vaccines were determined using the Mantel–Cox log rank test considering time to appearance of blood-stage parasites and the number of animals remaining uninfected. For all analyses, *p* values < 0.05 were considered statistically significant.

## 3. Results

### 3.1. Expression and Purification of rPfCSP-Based Vaccine Antigens

As diagrammed in Figure 1, four recombinant *Pf*CSP-based proteins were expressed and purified using an *E. coli* expression system. r*Pf*CSP (#1) consists of the N-terminal domain, 19 NVDP/NANP central repeats, and the C-terminal domain of *Pf*CSP. r*Pf*CSP/8 (#2) is a chimeric protein containing r*Pf*CSP (#1) fused to the r*Pf*MSP8 carrier protein. r*Pf*CSPN/8 (#3) contains the N-terminal domain and the first nine tetra-amino acid repeats of the central domain, fused to *Pf*MSP8. Finally, r*Pf*CSPC/8 (#4) consists of the last 10 tetra-amino acid repeats of the central domain followed by the C-terminal domain, fused to *Pf*MSP8. Recombinant proteins were expressed, purified by nickel chelate affinity chromatography, and analyzed by SDS-PAGE on 10% polyacrylamide gels run under both non-reducing and reducing conditions that were then stained with Coomassie blue (Figure 2A). Under both non-reducing and reducing conditions, r*Pf*CSP (#1) migrated as a doublet of approximately 38–39 kDa with several low molecular weight bands present in low concentration. The full-length product appeared larger than its predicted molecular weight. This is consistent with the aberrant migration of native *Pf*CSP, which has been attributed to the large, central NANP repeat domain. r*Pf*CSP/8 (#2), r*Pf*CSPN/8 (#3), and r*Pf*CSPC/8 (#4) migrated as distinct bands at ~76, ~61, and ~60 kDa, respectively, close to their predicted molecular weights.

Native *Pf*CSP contains two disulfide bonds in the C-terminal domain while the r*Pf*MSP8 carrier contains a C-terminal double EGF-like domain. The lack of high molecular weight aggregates on gels run under non-reducing conditions suggested that disulfide bonds in the r*Pf*CSP-based antigens formed correctly. To confirm the folding of the C-terminal domain of *Pf*CSP, vaccine antigens were separated by SDS-PAGE, run under non-reducing conditions, and analyzed by immunoblot with mAb 4B3, which recognizes a disulfide-dependent B cell epitope of *Pf*CSP. As shown in Figure 2B, mAb 4B3 was highly reactive with r*Pf*CSP (#1), r*Pf*CSP/8 (#2), and r*Pf*CSPC/8 (#4), which all contain the C-terminal domain of *Pf*CSP, but not with r*Pf*CSPN/8 (#3), which lacks this epitope. Comparable results were obtained with mAb 4C2 (not shown). Furthermore, reduction and alkylation of r*Pf*CSP (#1) to disrupt the folding of this domain eliminated reactivity with mAb 4B3. As such, folding of the r*Pf*CSP-based antigens mimicked the native protein and the folding of the C-terminal domain of *Pf*CSP was not altered by fusion to the *Pf*MSP8 carrier.

### 3.2. rPfCSP-Based Vaccines Induce High Titers of Anti-PfCSP IgG of Multiple Subclasses

Use of *Pf*MSP8 as a carrier has facilitated expression and purification of target antigens but has also provided additional T cell help to enhance antibody responses to protective, but subdominant B cell epitopes [34,36,37,38,39]. In Study #1, the immunogenicity of r*Pf*CSP (#1) and r*Pf*CSP/8 (#2) was compared to evaluate the impact of *Pf*MSP8 on the overall antibody response to near full-length r*Pf*CSP. In Study #2, the immunogenicity of r*Pf*CSPN/8 (#3) and r*Pf*CSPC/8 (#4) was compared to determine if the separation of the N-terminal and C-terminal domains positively or negatively impacts immunogenicity. The r*Pf*CSPC/8 (#4) is closest in design to the fusion protein present in the RTS,S and R21 *Pf*CSP-based vaccines but with the carrier protein switched from HBsAg to *Pf*MSP8. GLA-SE was selected as the adjuvant for these studies based on its overall robust performance when used to formulate r*Pf*MSP1/8, r*Pf*MSP2/8, and r*Pf*s25/8 vaccines [37,38]. Outbred CD1 mice (five males, five females per group) were immunized and boosted with 2.5 µg/dose of vaccine antigens at a 4-week interval.

Anti-*Pf*CSP and anti-*Pf*MSP8 antibody titers were determined for primary and secondary immunization sera of each animal by ELISA using r*Pf*CSP (#1)- or r*Pf*MSP8-coated plates. As expected, *Pf*CSP-specific antibody titers were increased significantly after boosting with r*Pf*CSP(#1) and r*Pf*CSP/8 (#2) (Figure 3A, Study #1) and with r*Pf*CSPN/8 (#3) and r*Pf*CSPC/8 (#4) (Figure 3A, Study #2). Analysis of secondary sera showed that immunization with r*Pf*CSP (#1) and r*Pf*CSP/8 (#2) elicited high titers of anti-*Pf*CSP-specific antibodies of comparable magnitude (Study #1, *p* = 0.25, ns). In contrast, two immunizations with r*Pf*CSPC/8 (#4) elicited significantly higher titers of anti-*Pf*CSP-specific antibodies in comparison to r*Pf*CSPN/8 (#3) (Study #2, *p* < 0.01). Secondary sera from all animals immunized with constructs containing the *Pf*MSP8 carrier contained high titers of anti-*Pf*MSP8-specific antibodies with significantly higher titers achieved by immunization with r*Pf*CSPC/8 (#4) vs. r*Pf*CSPN/8 (#3) (Figure 3A, Study #2, *p* < 0.01). Immunization with an admixture of r*Pf*CSPN/8 (#3) and r*Pf*CSPC/8 (#4) elicited high titers of anti-*Pf*CSP-specific antibodies comparable to immunization with r*Pf*CSPC/8 (#4) alone.

To investigate any potential sex-based differences in vaccine-induced immune responses, *Pf*CSP-specific secondary sera titers were stratified by sex (female, open symbols; male, closed symbols) within each immunization group. There were no significant differences in *Pf*CSP-specific antibody titers in secondary sera of female vs. male mice immunized with r*Pf*CSP (#1), r*Pf*CSP/8 (#2), or r*Pf*CSPN/8 (#3). However, female mice immunized with r*Pf*CSP/C (#4) had somewhat higher anti-*Pf*CSP-specific antibody titers than male mice in the same group (Figure 3A, Study #2, *p* = 0.02). These data indicate that all formulations were immunogenic; inducing high-titer, boostable anti-*Pf*CSP antibody responses; and that inclusion of r*Pf*MSP8 as carrier did not inhibit or increase the magnitude of the anti-*Pf*CSP antibody response to a nearly full-length construct (#1 vs. #2). With the truncated constructs, however, the immunogenicity of r*Pf*CSPC/8 (#4) was significantly greater than that of r*Pf*CSPN/8 (#3).

To begin to assess the quality of vaccine-induced antibody responses, the avidity and IgG subclass distribution of anti-*Pf*CSP-specific antibodies induced by each construct were determined. Avidity indices were calculated based on the resistance of antigen–antibody complexes to disruption by increasing concentrations of ammonium thiocyanate. As shown in Figure 3B (left panel), the avidity of anti-*Pf*CSP-specific antibodies induced by immunization with r*Pf*CSP (#1) vs. r*Pf*CSP/8 (#2) was comparable (Study #1, *p* = 0.11, ns). As shown in Figure 3B (right panel), there was a trend for slightly higher avidity antibodies induced by immunization with r*Pf*CSPC/8 (#4) in comparison to r*Pf*CSPN/8 (#3), but this did not reach statistical significance (Study #2, *p* = 0.065, ns).

Secondary sera from immunized mice were analyzed for production of IgG1, IgG2a/c, IgG2b, and IgG3 subtypes via ELISA using r*Pf*CSP (#1)-coated wells and quantitated against standard curves of myeloma proteins of each IgG subclass. In general, the subclass distribution reflected a balanced production of antigen-specific antibodies of the IgG1, IgG2a/c, and IgG2b subclasses with more limited production of IgG3 antibodies (Figure 3C). This is consistent with the production of both Th1 and Th2 cytokines characteristic of vaccines formulated with GLA-SE as adjuvant [47,48,49]. Mice immunized with r*Pf*CSP/8 (#2) produced slightly more IgG1 and IgG3 antibodies relative to those immunized with r*Pf*CSP (#1) (Figure 3C, Study #1). More notable was the induction of higher levels of anti-*Pf*CSP IgG2a/c antibodies by immunization with r*Pf*CSPC/8 (#4) in comparison to r*Pf*CSPN/8 (#3) (Figure 3C, Study #2, *p* < 0.01). Of additional interest, some of the variability in the subclass distribution of vaccine-induced antibodies in these outbred mice could be attributed to sex. Female mice immunized with r*Pf*CSPC/8 (#4) had higher levels of anti-*Pf*CSP-specific IgG1 and IgG2b antibodies than male mice in the same group (Figure 3C, Study #2, *p* < 0.02). While small differences were noted, the overall similarity in avidity and IgG subclass profiles suggest that anti-*Pf*CSP-specific antibodies induced by each of the four *Pf*CSP-based vaccine constructs were comparable in quality.

### 3.3. rPfCSP-Based Vaccines Induce T Cell Responses of Varied Specificity

In Immunization Study #3, outbred CD1 mice (five males, five females per group) were immunized and boosted as above with r*Pf*CSP(#1), r*Pf*CSP/8 (#2), r*Pf*CSPN/8 (#3), or r*Pf*CSPC/8 (#4) formulated with GLA-SE as adjuvant, or with GLA-SE alone. Analysis of vaccine-induced anti-*Pf*CSP antibody responses confirmed the results of Study #1 and Study #2. Immunization with r*Pf*CSP (#1), r*Pf*CSP/8 (#2), and r*Pf*CSPC/8 (#4) elicited high and comparable titers of *Pf*CSP-specific IgG while the response induced by the r*Pf*CSPN/8 (#3) vaccine construct was significantly lower (Appendix A, panel A). As antigen-specific T cell responses have been shown to contribute to protection against pre-erythrocytic stage malaria parasites [50], the magnitude and specificity of vaccine-induced T cell responses was measured, using antigen-specific IFNγ production as a readout. Splenocytes were harvested from mice immunized with each of the four vaccine constructs and stimulated with r*Pf*CSP (#1) or r*Pf*MSP8, representing the constituent domains of the chimeric vaccines, or with pools of 15-mer overlapping synthetic peptides spanning the central repeat region and C-terminal domains of *Pf*CSP. IFNγ released into culture supernatants was quantified by ELISA.

Immunization with r*Pf*CSP (#1) and r*Pf*CSPC/8 (#4) elicited *Pf*CSP-specific T cell responses, consistent with the presence of known CD4+ T cell epitopes in the C-terminal domain of *Pf*CSP [51,52] (Figure 4A). This was confirmed upon stimulation of splenocytes with a pool of six *Pf*CSP C-terminal peptides (Figure 4B, Appendix A, peptides #31, 33–37). In both assays, the range in concentration of IFNγ secreted was fairly large, likely reflecting the genetic heterogeneity of outbred CD1 mice. As expected, *Pf*CSP-specific T cell responses elicited by immunization with r*Pf*CSPN/8 (#3) were minimal, due to the absence of the C-terminal *Pf*CSP T cell epitopes in this construct. Somewhat unexpected was the rather low *Pf*CSP-specific T cell responses elicited by immunization with r*Pf*CSP/8 (#2), which contains all of the same *Pf*CSP T cell epitopes found in r*Pf*CSP (#1) and r*Pf*CSPC/8 (#4). However, immunization with r*Pf*CSP/8 (#2), as well as r*Pf*CSPN/8 (#3) and r*Pf*CSPC/8 (#4), elicited strong T cell responses to the r*Pf*MSP8 carrier that were relatively consistent among individual animals across the three groups (Figure 4C). There were no significant sex-related differences in antigen-specific T cell responses with the exception that female mice immunized with r*Pf*CSPN/8 (#3) responded to r*Pf*MSP8 better than male mice (*p* = 0.03). These data show that immunization with the r*Pf*CSP-based vaccines induce strong T cell responses, but that fusion of nearly full-length *Pf*CSP to *Pf*MSP8 shifted the specificity of the response away from *Pf*CSP-specific epitopes resulting in a more dominant *Pf*MSP8-specific response.

### 3.4. Immunization with Chimeric rPfCSP/8 (#2) Protects against Sporozoite Challenge Infection

In Immunization Study #4, the efficacy of the four r*Pf*CSP-based vaccines was assessed. Outbred CD1 mice (five males, five females per group) were immunized and boosted as above with r*Pf*CSP (#1), r*Pf*CSP/8 (#2), r*Pf*CSPN/8 (#3), and r*Pf*CSPC/8 (#4) formulated with GLA-SE as adjuvant, or with GLA-SE alone. Furthermore, a group of mice immunized with the r*Pf*MSP8 carrier only, formulated with GLA-SE as adjuvant, was added to the efficacy study to assess any carrier-dependent effects. Analysis of pre-challenge antibody responses again demonstrated reproducibly high levels of anti-*Pf*CSP-specific IgG with three of the four *Pf*CSP-based vaccine constructs (Appendix A, panel B). Three weeks following the second immunization, mice were infected by i.v. injection of 6250 transgenic *P. yoelii* 17X sporozoites engineered to express *P. falciparum* CSP from the endogenous locus [46]. Sterile protection was assessed by monitoring Giemsa-stained thin tail-blood smears for the presence of blood-stage parasites for 14 days.

As shown in Figure 5 and Table 1, control mice immunized with GLA-SE alone (9/10) or with the recombinant *Pf*MSP8 carrier (9/10) developed blood-stage malaria within the first 9 days post-challenge. Mice immunized with r*Pf*CSP/8 (#2) showed high and significant protection compared to the control groups, with sterile protection in seven out of nine animals. Despite the induction of high titers of anti-*Pf*CSP antibodies, sterile protection in mice immunized with r*Pf*CSP (#1) or with the truncated r*Pf*CSPC/8 (#4) was limited and not significantly different than controls. No significant protection was observed in animals immunized with r*Pf*CSPN/8 (#3), consistent with lower overall *Pf*CSP-specific T and B cell responses. These data suggest that in addition to the major NANP containing repeat domain and the C-terminus of *Pf*CSP, inclusion of the N-terminal domain, junctional region, and/or minor repeat domains can improve vaccine efficacy. Although the enhanced efficacy depended on fusion to the *Pf*MSP8 carrier, the protection could not be correlated with anti-*Pf*CSP specific antibody titers.

### 3.5. Immunization with PfCSP-Based Vaccines Drives a Strong Antibody Response to Conformational B Cell Epitopes

The C-terminal domain of *Pf*CSP contains two disulfide bonds. mAbs that recognize conformation-dependent epitopes in this C-terminal domain of *Pf*CSP can neutralize *P. falciparum* sporozoites. To assess the induction of antibodies to linear versus conformational epitopes of *Pf*CSP, sera from mice immunized with each of the four r*Pf*CSP-based vaccines were concurrently assayed by ELISA on wells coated with r*Pf*CSP (#1) or fully reduced and alkylated (R/A) r*Pf*CSP (#1) that lacked disulfide-dependent B cell epitopes. The proportion of antibodies in each sera recognizing linear B cell epitopes was expressed as the ratio of the titer on R/A r*Pf*CSP (#1) divided by the titer on intact r*Pf*CSP (#1). Ratios using sera collected from Immunization Studies #1 through #4 are summarized in Figure 6 (*n* = 27–30/group).

As shown in Figure 6, animals immunized with r*Pf*CSPN/8 (#3) produced antibodies that recognized linear epitopes of *Pf*CSP with the R/A r*Pf*CSP to r*Pf*CSP antibody response ratio near 1 (0.94 ± 0.18). This was as expected, as r*Pf*CSPN/8 (#3) lacks the C-terminal, conformational domain of *Pf*CSP. Unexpectedly, a considerable proportion of the antibody responses induced by immunization with r*Pf*CSP (#1), r*Pf*CSP/8 (#2), and r*Pf*CSPC/8 (#4) targeted conformational B cell epitopes. The mean R/A r*Pf*CSP to r*Pf*CSP antibody response ratio in r*Pf*CSP (#1)-immunized animals was 0.41 ± 0.14 and comparable to ratios observed in r*Pf*CSP/8 (#2)- and r*Pf*CSPC/8 (#4)-immunized mice (0.48 ± 0.12 and 0.38 ± 0.18, respectively). As such, fusion of *Pf*CSP sequences to the *Pf*MSP8 carrier did not impact antibody responsiveness to the C-terminus of *Pf*CSP. Stratifying these R/A *Pf*CSP to *Pf*CSP ratios by sex revealed that there were no significant differences between males and females within each group. Overall, these data indicate that approximately half of the antibodies induced by three of four r*Pf*CSP-based vaccine constructs recognize disulfide-dependent epitopes in the C-terminal domain of *Pf*CSP.

### 3.6. Variability in the Recognition of Linear B Cell Epitopes by Antibodies Induced by Immunization with PfCSP-Based Vaccines

To assess the fine specificity of vaccine-induced antibodies against linear B cell epitopes of *Pf*CSP, ELISAs were conducted using 37 overlapping 15-mer biotinylated peptides, spanning the length of the r*Pf*CSP (#1) construct (Appendix A). For the initial screen, pools of sera from male and female mice immunized with r*Pf*CSP (#1), r*Pf*CSP/8 (#2), r*Pf*CSPN/8 (#3), r*Pf*CSPC/8 (#4), or GLA-SE were screened for reactivity with each peptide. As shown in Appendix A, multiple peptides from the N-terminal junctional region, minor repeat region, the central repeat domain, and to a lesser degree, the C-terminal domain were recognized by sera from immunized mice. Based on these data, a set of 13 informative peptides (Figure 7A) were selected to investigate the diversity of the anti-*Pf*CSP antibody responses in individual sera obtained from Immunization Studies #1 through #4 (*n* = 27–30/group).

**Figure 7 vaccines-12-00351-f007:**
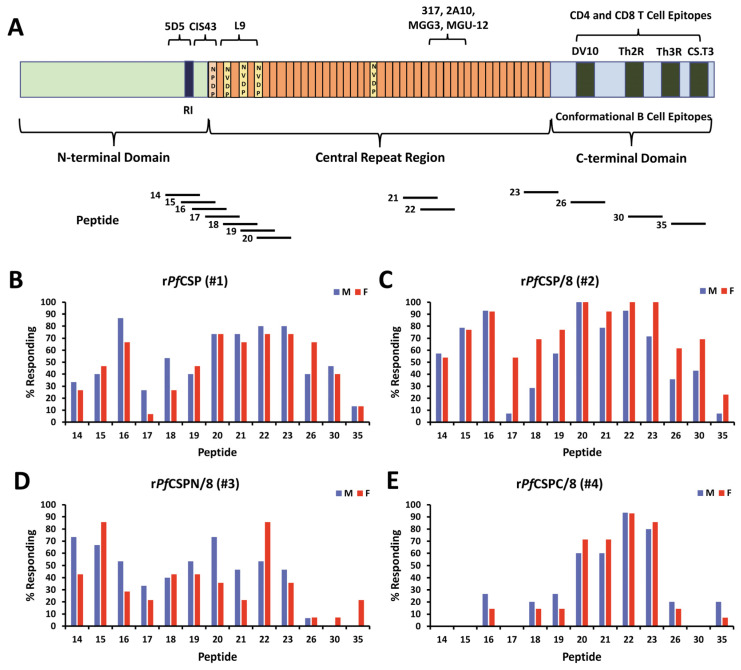
Fine specificity of vaccine-induced antibodies against linear B cell epitopes of *Pf*CSP. (**A**) Schematic of *Pf*CSP (adapted from Ref. [18]) showing the approximate location of 13 synthetic peptides within the N-terminal domain, central repeat region, and C-terminal domain. Numbering is consistent with that shown in Appendix A. Mapped B cell epitopes recognized by known neutralizing antibodies, as well as C-terminal CD4^+^ and CD8^+^ T cell epitopes, are indicated at the top of the figure. The epitope specificity of antibodies from mice immunized with (**B**) r*Pf*CSP (#1), (**C**) r*Pf*CSP/8 (#2), (**D**) r*Pf*CSPN/8 (#3), or (**E**) r*Pf*CSPC/8 (#4) (Immunization Studies #1–#4, *n* = 27–30 animals per group; male (blue bars), female (red bars)) were determined by ELISA on the indicated peptides. Sera, assayed at a dilution of 1:1000, were considered reactive if the mean OD_405_ of test sera on a given peptide was greater than mean OD_405_ + 3 standard deviations of GLA-SE control sera. Values are expressed as the percent of animals responding to each peptide (y-axis).

r*Pf*CSPC/8 (#4) is the construct most similar to RTS,S and R21 vaccines. As shown in Figure 7E, r*Pf*CSPC/8 (#4) induced antibody responses that bound NANP repeat containing peptides in 60–90% of animals (i.e., peptides 20, 21, 22, 23). The limited recognition of peptides derived from the C-terminal regions was expected, given the conformational nature of this domain. As shown in Figure 7D, immunization with r*Pf*CSPN/8 (#3) shifted the response away from NANP-containing epitopes to some degree with increased recognition of linear epitopes associated with the junctional region and minor-repeat regions (i.e., peptides 14, 15, 16, 18, 19). Importantly, immunization with the near full-length r*Pf*CSP (#1) increased the breadth of linear B cell epitopes recognized to include peptides derived from the N-terminal domain, the junctional region, the minor NVDP repeat regions, as well as the central NANP repeat domain (Figure 7B). Fusion of the nearly full-length *Pf*CSP sequence to the *Pf*MSP8 carrier did not further increase the diversity of epitopes recognized. However, r*Pf*CSPC/8 (#4) immunization increased the overall percentage of animals seroreactive to these linear epitopes with 70–100% of animals recognizing multiple peptides.

To quantitate differences in breadth of linear B cell epitopes recognized, the number of peptides each animal recognized was determined. Values were then compared across the four immunization groups. As shown in Figure 6B, animals immunized with r*Pf*CSP/8 (#2) recognized a significantly greater number of peptides than animals immunized with either r*Pf*CSPN/8 (#3) or r*Pf*CSPC/8 (#4). While the mean number of peptides recognized by animals immunized with r*Pf*CSP/8 (#2) was elevated compared to animals immunized with r*Pf*CSP (#1), the difference was not statistically significant. Combined, these data show that addition of the N-terminal domain, junctional region, and minor-repeat regions to the RTS,S-like r*Pf*CSPC/8 (#4) construct expands the breadth of antibody responses to include novel, potentially protective, linear B cell epitopes.

### 3.7. Antibody Recognition of Disulfide Bond-Dependent Epitopes of PfCSP Are Not Required for PfCSP-Based Vaccine-Induced Protection

The magnitude of the antibody response to conformation-dependent epitopes elicited by immunization with r*Pf*CSP (#1) or r*Pf*CSP/8 (#2) was striking. In Immunization Study #5, the contribution of these antibodies to vaccine-induced protection was evaluated. Outbred CD1 mice (five males, five females per group) were immunized and boosted with r*Pf*CSP (#1) and r*Pf*CSP/8 (#2) as above or with r*Pf*CSP (#1) and r*Pf*CSP/8 (#2) that had been reduced and alkylated to disrupt disulfide bonds required for proper conformation of the C-terminus of *Pf*CSP. All vaccines were formulated with GLA-SE as adjuvant. As expected, immunization with r*Pf*CSP (#1) or r*Pf*CSP/8 (#2) elicited high titers of anti-*Pf*CSP antibodies (Figure 8A), a sizable proportion of which recognized conformational epitopes (Figure 8B). In contrast, anti-*Pf*CSP antibody responses induced by immunization with R/A r*Pf*CSP (#1) or with R/A r*Pf*CSP/8 (#2) were decreased relative to the corresponding folded antigens (Figure 8A), and highly restricted to recognition of linear determinants (Figure 8B). To assess impact on efficacy, immunized mice were then infected with transgenic *P. yoelii* 17X sporozoites expressing *P. falciparum* CSP and sterile protection monitored as above. As shown in Figure 8 and Table 2, the lack of antibodies directed against C-terminal conformational B cell epitopes of *Pf*CSP did not alter vaccine-induced protection.

### 3.8. Immunization with PfCSP-Based Vaccines Drives Strong, Durable Antibody Responses

In the final Immunization Study #6, the durability of the antibody response induced by the four r*Pf*CSP-based vaccines was evaluated. Outbred CD1 mice (five males, five females per group) were immunized and boosted as above with r*Pf*CSP (#1), r*Pf*CSP/8 (#2), r*Pf*CSPN/8 (#3), or r*Pf*CSPC/8 (#4). Following the boost, serum samples were collected once per month for a period of 6 months and anti-*Pf*CSP antibody titers in each test bleed were determined by ELISA. Figure 9A shows that immunization with three of the four constructs induced high titers of anti-*Pf*CSP antibodies. As expected, the anti-CSP antibody titers dropped significantly from month one (M1) to month six (M6). However, except for animals immunized with r*Pf*CSPN/8 (#3), the overall anti-*Pf*CSP titer was relatively stable, only dropping by ~twofold during the 6 month follow up period. In contrast, anti-*Pf*CSP antibodies induced by r*Pf*CSPN/8 (#3) immunization dropped markedly, with ~sixfold reduction in titer over the 6 month period. To determine if there was any preferential drop in antibodies that recognized linear versus conformational epitopes, the R/A r*Pf*CSP to r*Pf*CSP antibody response ratio was again determined. Mice immunized with r*Pf*CSPC/8 (#4) had a small but significant reduction in the R/A r*Pf*CSP vs. r*Pf*CSP response ratio when comparing M1 to M6, indicative of a preferential drop in antibodies directed toward linear epitopes (Figure 9B). However, this was not observed in animals immunized with either r*Pf*CSP (#1) or r*Pf*CSP/8 (#2). Overall, these data show that three of the four *Pf*CSP-based vaccines induce high and durable anti-*Pf*CSP antibody responses in mice.

## 4. Discussion

The goal of this study was to consider changes in the design of the pre-erythrocytic-stage RTS,S [19,21] and R21 [20] malaria vaccines that could potentially improve efficacy. Both of these vaccines target the central repeat and C-terminal domains of *Pf*CSP and induce antibodies to the central repeat, which are critical for protection [9,14,15]. With this in mind, we first evaluated novel *Pf*CSP-based constructs designed to include epitopes outside of major NANP repeat epitope in an effort to improve the diversity of B cell epitopes recognized. Second, we evaluated a promising malaria-specific carrier protein, *Pf*MSP8, as a substitute for the heterologous HBsAg carrier. In general, we focused our comparisons on the immunogenicity and efficacy of r*Pf*CSP (#1), r*Pf*CSP/8 (#2), and r*Pf*CSPN/8 (#3) relative to r*Pf*CSPC/8 (#4), which is most similar in design to R21 and the *Pf*CSP-specific component of RTS,S. With all four constructs, the avidity and IgG subclass profiles of vaccine-induced antibodies were similar and may reflect the use of GLA-SE as adjuvant for all formulations [37,38,47,48,49]. We noted differences in the magnitude, specificity, durability, and efficacy of vaccine-induced responses. For the most part, these differences in immunogenicity and efficacy were independent of sex and directly related to construct design.

We achieved success in increasing the diversity of B cell epitopes recognized upon immunization by inclusion of the N-terminal domain, the junctional region, and the minor repeats with r*Pf*CSP (#1) and r*Pf*CSP/8 (#2). Antibodies from most animals immunized with r*Pf*CSP (#1), r*Pf*CSP/8 (#2), and r*Pf*CSPC/8 (#4) strongly recognized peptides containing the immunodominant NANP repeat epitope. In addition, animals immunized with near full-length r*Pf*CSP (#1) and r*Pf*CSP/8 (#2) also recognized linear epitopes located near the junction of the N-terminal domain and central repeat region, which included peptides containing the minor NVDP repeats. Although the fine specificity of these antibodies elicited in mice may differ from neutralizing human mAbs such as CIS43 [22] and L9 [24], the data show that this region of *Pf*CSP is immunogenic in r*Pf*CSP (#1) and r*Pf*CSP/8 (#2). These epitopes were similarly recognized by mice immunized with r*Pf*CSPN/8 (#3), which includes the same N-terminal domain and junctional region. However, the response to r*Pf*CSPN/8 (#3) immunization was associated with a somewhat less consistent antibody response to the major NANP repeat epitope, suggesting that the inclusion of the C-terminal domain of *Pf*CSP may be beneficial for overall immunogenicity. The C-terminal domain of *Pf*CSP also contains two disulfide bonds and mAbs that recognize conformational, C-terminal epitopes neutralize *P. falciparum* sporozoites [42,53]. In our hands, the proportion of vaccine-induced antibodies that recognized this conformationally constrained domain was expectedly high in animals immunized with r*Pf*CSP (#1), r*Pf*CSP/8 (#2), and r*Pf*CSPC/8 (#4). This was demonstrated by a drop of ~50% in ELISA titers when assayed using reduced and alkylated r*Pf*CSP (#1) vs. intact r*Pf*CSP (#1) as the coating antigen (R/A *Pf*CSP-to-*Pf*CSP response ratio). Additionally, there was minimal recognition of linear peptide epitopes in this C-terminal region of *Pf*CSP across groups.

In addition to providing B cell help to enhance both the magnitude and the quality of vaccine-induced antibodies, *Pf*CSP-specific T cells can independently contribute to protection. Using antigen-specific IFNγ production as a metric of responsiveness, we showed that immunization with both r*Pf*CSP (#1)- and r*Pf*CSPC/8 (#4)-induced T cells that recognized the C-terminal domain of *Pf*CSP, which contains well-characterized T cell epitopes [21,51,52,54]. The *Pf*CSP-specific T cell response in mice immunized with r*Pf*CSPN/8 (#3) was absent, consistent with the lack of this C-terminal domain in this construct. Unexpectedly, immunization r*Pf*CSP/8 (#2), which contains the C-terminal T cell epitopes of *Pf*CSP, did not induce *Pf*CSP-specific T cell responses. All constructs containing the *Pf*MSP8 carrier induced a consistent *Pf*MSP8-specific T cell response, similar to our previous studies with blood-stage and sexual-stage chimeric vaccines [33,34,37,38]. These results suggest that fusion of a nearly full-length *Pf*CSP sequence to the *Pf*MSP8 carrier shifted the T cell response away from *Pf*CSP-specific epitopes toward *Pf*MSP8 in the r*Pf*CSP/8 (#2) vaccine.

Immunization with the r*Pf*CSP fused to *Pf*MSP8 allows for cognate recognition of T and B cell epitopes in a single protein upon immunization. Combined, our data indicate that T cell recognition of strong epitopes within the *Pf*MSP8 carrier is effective in providing help to *Pf*CSP-specific B cells, driving strong, class-switched, durable B cell/Ab responses. We previously demonstrated this with chimeric MSP1/8 vaccines in both *P. yoelii* and *P. falciparum*. The ability of *Pf*MSP8-specific T cells to provide help to *Pf*CSP-specific B cells combined with the use of the larger, nearly full-length *Pf*CSP component in *Pf*CSP/8 (#2) partially explains the increase in diversity of the B cell response. In addition, we previously showed in studies with *Pf*MSP2, which is known to form amyloid-like fibrils [55,56,57], that fusion to the *Pf*MSP8 carrier prevents fibril formation and the masking of B cell epitopes in the *Pf*MSP2 domain. This increased the induction of *Pf*MSP2-specific antibodies of greater diversity which were functional in the opsonophagocytosis of merozoites [36,37]. A similar influence on the conformation of *Pf*CSP when fused to *Pf*MSP8 may also have contributed to an increase in the breadth of the B cell response.

We designed our efficacy studies to accentuate potential differences in protection when comparing our *Pf*CSP-based vaccines by including only two immunizations (2.5 µg/dose) followed by a robust sporozoite challenge with transgenic *P. yoelii* sporozoites expressing *Pf*CSP from the endogenous locus [24]. The r*Pf*CSP/8 (#2) vaccine provided a consistently high level of sterile protection against sporozoite challenge. This was in contrast to the corresponding r*Pf*CSP (#1) lacking the *Pf*MSP8 carrier, in which protection was partial, trending toward statistical significance. This may reflect reduced variability in vaccine-induced antibody responses in these outbred mice immunized with r*Pf*CSP/8 (#2) in comparison to r*Pf*CSP (#1). Considering our T cell data, these results also indicate that T cell responses to epitopes in the C-terminus of *Pf*CSP, which were lacking in r*Pf*CSP/8 (#2)-immunized mice, are not required for protection induced by this construct. By inclusion of an additional group of animals immunized with the *Pf*MSP8 carrier alone, we demonstrated that protection is dependent on the *Pf*CSP-specific effector responses, and not the effector activity of antibodies or T cells specific for *Pf*MSP8. Protection induced by immunization with r*Pf*CSPN/8 (#3) and r*Pf*CSPC/8 (#4), which lack a full complement of potentially protective B cell epitopes, was not significantly different than r*Pf*MSP8 and adjuvant alone control groups.

The reduction and alkylation of r*Pf*CSP (#1) and r*Pf*CSP/8 (#2) impacted immunogenicity, in part due to the loss of antibodies that recognize C-terminal, conformation dependent epitopes. Unexpectedly, the loss of this rather large population of *Pf*CSP-specific antibodies did not impact protection. At least two explanations seem plausible. First, the vaccine-induced anti-NANP repeat antibody response alone may be of sufficient magnitude to confer significant protection in this vaccine model system. Mounting data from phase II and III clinical trials with the R21 *Pf*CSP vaccine, lacking the N-terminal and junctional region epitopes, identified the strength of the anti-NANP repeat antibody response as a correlate of protection [14,15,16]. However, this does not exclude the possibility that under circumstances when a less robust, NANP-specific response is elicited by immunization, antibodies to conformational epitopes in the C-terminus of *Pf*CSP may contribute to protection [58]. This possibility is certainly supported by studies with mAbs [42,53] and provides a basis for the concept of passive immunization with *Pf*CSP-specific mAbs to non-NANP epitopes to protect against seasonal malaria [59,60,61]. Alternatively, it has been reported that in vivo, the N-terminal domain of *Pf*CSP folds to mask the C-terminal, TSR-containing adhesive domain of *Pf*CSP, to facilitate exit of sporozoites from the dermis following a mosquito bite [27,28]. In this in vivo setting, conformation-dependent C-terminal epitopes of *Pf*CSP on the surface of *P. falciparum* sporozoites may not be accessible for antibody binding, allowing for parasites to escape neutralization as they transit from the skin to the liver. This may in part have contributed to our inability to establish clear correlates of *Pf*CSP vaccine-induced, polyclonal antibody-mediated protection in this study.

Our data, considered in the framework of our immunization protocols and parameters, clearly show added value of *Pf*CSP-based vaccines designed to include the N-terminal domain, the central repeat region, and the C-terminal domain, with respect to maintaining a strong anti-repeat antibody response, while driving responses to additional, potentially protective B cell epitopes. Inclusion of *Pf*MSP8 as a vaccine carrier facilitated expression and purification but most importantly improved efficacy. Based on the data with our near full-length *Pf*CSP constructs (#1, #2), a side-by-side comparison of *Pf*MSP8 and HBsAg as carrier may be of interest. However, we anticipate further benefit from the inclusion of *Pf*MSP8 as a carrier when formulating a *Pf*CSP-based vaccine in combination with other targets, which we firmly believe will be required for success. In this regard, we have demonstrated that immunization with r*Pf*MSP1, r*Pf*MSP2, and r*Pf*s25 vaccines required fusion of the domain targeted by each vaccine to *Pf*MSP8 to maintain the immunogenicity of each component when formulated in a trivalent combination. Effectively adding r*Pf*CSP/8 (#2) to this formulation has potential to achieve the goal of providing strong protection against pre-erythrocytic-stage parasites concurrent with the ability to neutralize breakthrough blood-stage parasites and protect children from life-threatening malaria.

## Figures and Tables

**Figure 1 vaccines-12-00351-f001:**
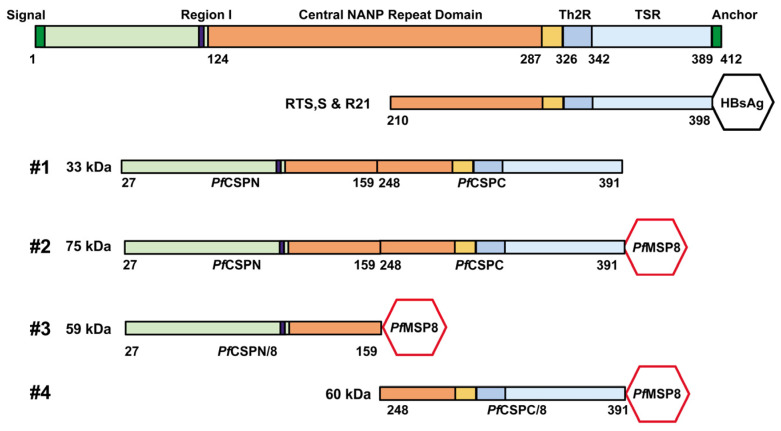
Schematic of four novel recombinant *Pf*CSP-based vaccines constructs. **Top**: Full-length *Pf*CSP including signal peptide, Region I, central NANP repeat domain, the Th2R CD4^+^ T cell epitope, thrombospondin-like type I repeat (TSR) domain, and C-terminal glycosylphosphatidylinositol (GPI) anchor sequence. Amino acid locations are indicated below the image. **Below**: RTS,S and R21 HBsAg fusion proteins; (#1) r*Pf*CSP containing the N-terminus, 19 central repeats, and the C-terminus; (#2) r*Pf*CSP/8 comprising r*Pf*CSP (#1) fused to r*Pf*MSP8; (#3) r*Pf*CSPN/8) containing the N-terminus and the first 9 tetra-amino acids repeats of *Pf*CSP fused to *Pf*MSP8; (#4) *Pf*CSPC/8 containing 10 NANP repeats and the C-terminus of *Pf*CSP fused to *Pf*MSP8.

**Figure 2 vaccines-12-00351-f002:**
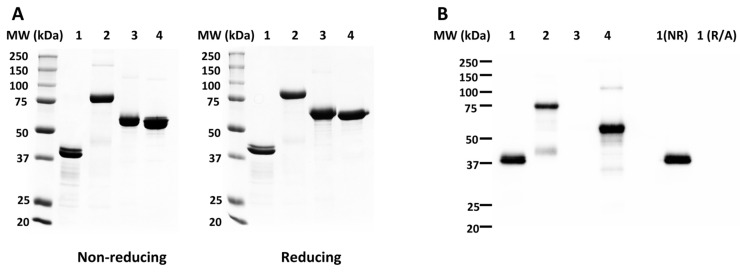
Assessment of purified r*Pf*CSP-based vaccines. (**A**) Purified r*Pf*CSP-based vaccines (#1–#4) were separated by SDS-PAGE (10% gel, 3 µg/lane) under non-reducing and reducing conditions, followed Coomassie blue staining. (**B**) Purified, non-reduced r*Pf*CSP-based vaccines (#1–#4) were separated by SDS-PAGE (10% gel, 0.5 µg/lane) followed by immunoblot analysis using mAb 4B3, which recognizes a C-terminal conformational epitope of *Pf*CSP. Control lanes include non-reduced r*Pf*CSP (#1) (NR) vs. fully reduced and alkylated r*Pf*CSP (#1) (R/A).

**Figure 3 vaccines-12-00351-f003:**
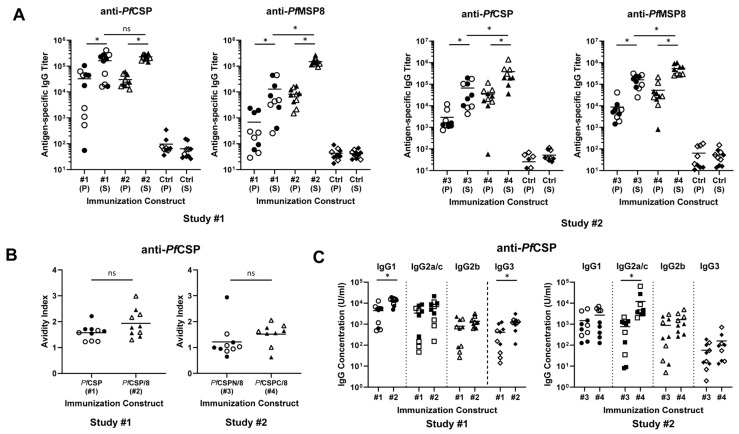
Magnitude and quality of vaccine-induced antibody responses. Groups of CD1 mice (*n* = 9–10/group) (female, open symbols; males, closed symbols) were immunized and boosted with r*Pf*CSP (#1, circles) or r*Pf*CSP/8 (#2, triangles) in Study #1 or r*Pf*CSPN/8 (#3, circles) or r*Pf*CSPC/8 (#4, triangles) in Study #2. Control animals received GLA-SE alone (diamonds). (**A**) Using primary (P) and secondary (S) immunization sera, antigen-specific IgG titers were determined by ELISA using wells coated with r*Pf*CSP (#1) or r*Pf*MSP8. (**B**) Avidity and (**C**) IgG subclass profiles (IgG1—circles; IgG2a/c—squares; IgG2b—triangles; IgG3—diamonds) using secondary sera obtained from animals in Study #1 and #2 were determined by ELISA using wells coated with r*Pf*CSP (#1). Symbols depict measurements from individual animals. Statistical significance of (i) differences in titer between primary and secondary sera were assessed using the Wilcoxon matched pairs signed-rank test and (ii) differences in titer, avidity index, and IgG subclass concentration between two groups were assessed by Mann–Whitney test. * *p* < 0.05; ns: non-significant.

**Figure 4 vaccines-12-00351-f004:**
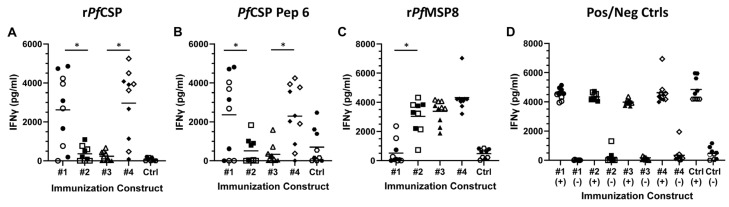
Magnitude and specificity of T cell responses induced by immunization with *Pf*CSP-based vaccines. Groups of CD1 mice (*n* = 9–10/group) (female, open symbols; male, closed symbols) were immunized and boosted with r*Pf*CSP (#1, circles), r*Pf*CSP/8 (#2, squares), r*Pf*CSPN/8 (#3, triangles), or r*Pf*CSPC/8 (#4, diamonds), or with adjuvant alone (Ctrl, hexagons). Splenocytes were stimulated with (**A**) r*Pf*CSP (#1) (10 µg/mL), (**B**) a pool of six 15-residue C-terminal *Pf*CSP peptides (3 µg/mL of each peptide), or (**C**) r*Pf*MSP8 (10 µg/mL). (**D**) Concanavalin A (1 µg/mL) and cells alone were used as positive (+) and negative (−) controls, respectively. The concentration of IFNγ released into culture supernatants (pg/mL) was measured by ELISA. Statistical significance of the differences in IFNγ production between r*Pf*CSP (#1) vs. r*Pf*CSP/8 (#2) groups and between r*Pf*CSPN/8 (#3) vs. r*Pf*CSPC/8 (#4) groups were assessed by Mann–Whitney test. * *p* < 0.05.

**Figure 5 vaccines-12-00351-f005:**
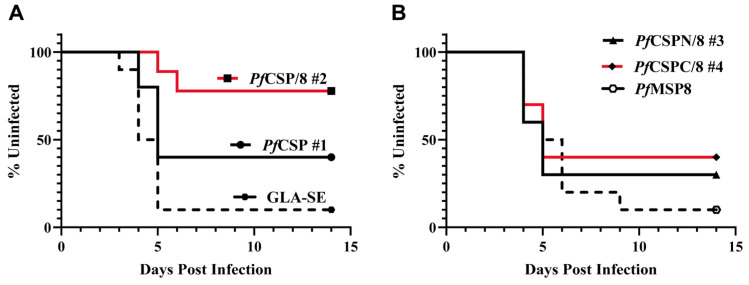
Efficacy of *Pf*CSP-based vaccines. Groups of CD1 mice (*n* = 9–10/group, both sexes) were immunized and boosted with (**A**) r*Pf*CSP (#1), r*Pf*CSP/8 (#2), or GLA-SE alone and (**B**) r*Pf*CSPN/8 (#3), r*Pf*CSPC/8 (#4), or r*Pf*MSP8. Three weeks following the second immunization, animals were challenged i.v. with 6250 transgenic *P. yoelii* 17X sporozoites expressing *P. falciparum* CSP. Sterile protection was monitored using Giemsa-stained thin tail-blood smears to detect blood-stage parasites for 14 days. Kaplan–Meyer curves represent time to appearance of blood-stage parasites (x-axis) and percentage of animals remaining uninfected (y-axis).

**Figure 6 vaccines-12-00351-f006:**
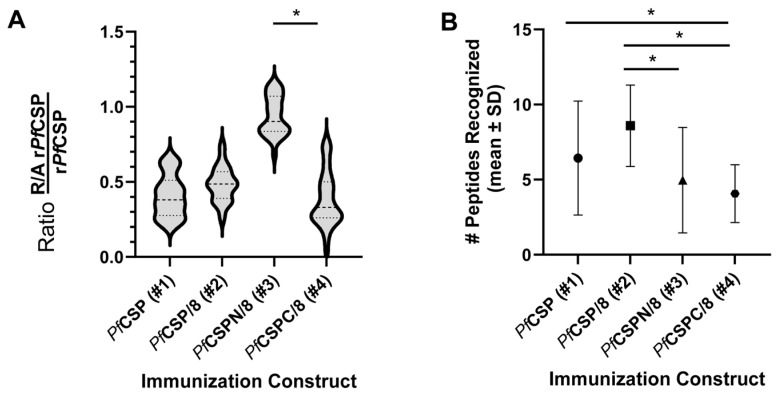
(**A**). Recognition of linear versus conformational epitopes of *Pf*CSP. Secondary sera from mice immunized with r*Pf*CSP (#1), r*Pf*CSP/8 (#2), r*Pf*CSPN/8 (#3), or r*Pf*CSPC/8 (#4) were assayed by ELISA on plates coated with r*Pf*CSP (#1) or R/A r*Pf*CSP (#1) (Immunization studies #1–#4, *n*= 27–30 animals per group, both sexes). The proportion of antibodies in each sera recognizing linear B cell epitopes was expressed as the ratio of the titer on R/A r*Pf*CSP (#1) divided by the titer on intact r*Pf*CSP (#1). Violin plots depict group mean (± SD, dashed lines) with statistical significance of differences in ratios assessed using the Kruskal–Wallis test followed by Dunn’s correction for multiple comparisons. * *p* < 0.05. (**B**) Breadth of linear B cell epitopes recognized by immunized animals. Secondary sera from mice immunized with r*Pf*CSP (#1, circle), r*Pf*CSP/8 (#2, square), r*Pf*CSPN/8 (#3, triangle), or r*Pf*CSPC/8 (#4, hexagon) (Immunization Studies #1–#4, *n* = 27–30 animals per group, both sexes) were evaluated by ELISA for reactivity with 13 individual *Pf*CSP peptides (see detailed data in Figure 7). Sera, assayed at a dilution of 1:1000, were considered reactive (# peptides recognized, y-axis) if the mean OD_405_ of test sera on a given peptide was greater than mean OD_405_ + 3 standard deviations of GLA-SE control sera. Significance of differences in mean number (±SD) of peptides recognized by mice in each immunization group was assessed using the Kruskal–Wallis test followed by Dunn’s correction for multiple comparisons. * *p* < 0.05.

**Figure 8 vaccines-12-00351-f008:**
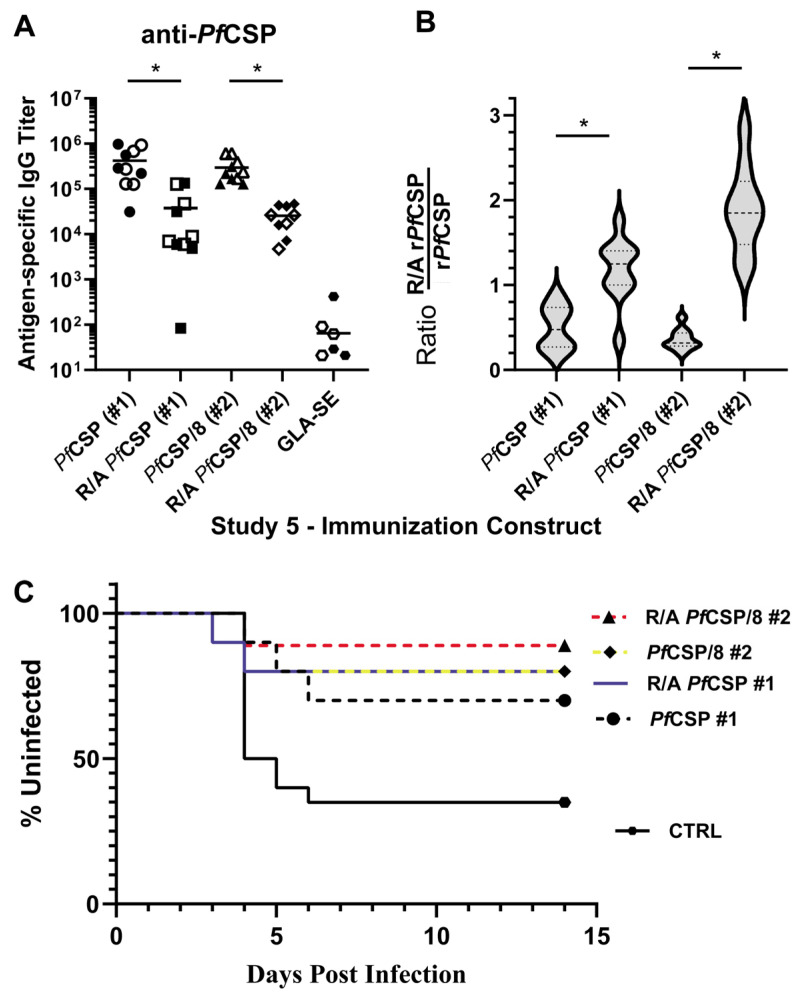
Contribution of antibodies to conformation-dependent epitopes of *Pf*CSP to vaccine-induced protection. (**A**) Groups of CD1 mice (*n* = 9–10/group, both sexes) were immunized and boosted with r*Pf*CSP (#1, circles), r*Pf*CSP/8 (#2, triangles), or fully reduced and alkylated R/A r*Pf*CSP (#1, squares) or R/A r*Pf*CSP/8 (#2, diamonds). Animals immunized with r*Pf*MSP8 or GLA-SE alone served as negative controls (hexagon). Antigen-specific IgG titers in secondary sera were measured by ELISA on wells coated with r*Pf*CSP (#1). (**B**) The proportion of antibodies in each sera recognizing linear B cell epitopes was expressed as the ratio of the titer on R/A r*Pf*CSP (#1) divided by the titer on intact r*Pf*CSP (#1) (y-axis). Violin plots depict group mean (±SD, dashed lines). Statistical significances of difference in (**A**,**B**) were evaluated using the Kruskal–Wallis test followed by Dunn’s correction for multiple comparisons. * *p* < 0.05. (**C**) Vaccine efficacy was determined upon challenge infection with 6250 transgenic *P. yoelii* 17X sporozoites expressing *P. falciparum* CSP. Sterile protection was monitored using Giemsa-stained thin tail-blood smears to detect the presence of blood-stage parasites for 14 days. Kaplan–Meyer curves represent the time to appearance of blood-stage parasites (x-axis) and the percentage of animals remaining uninfected (y-axis). *Pf*CSP (#1)—black dashed line; *Pf*CSP/8 (#2)—yellow dashed line; R/A r*Pf*CSP (#1)—blue solid line; R/A r*Pf*CSP/8 (#2)—red dashed line; r*Pf*MSP8 + GLA-SE controls—black solid line.

**Figure 9 vaccines-12-00351-f009:**
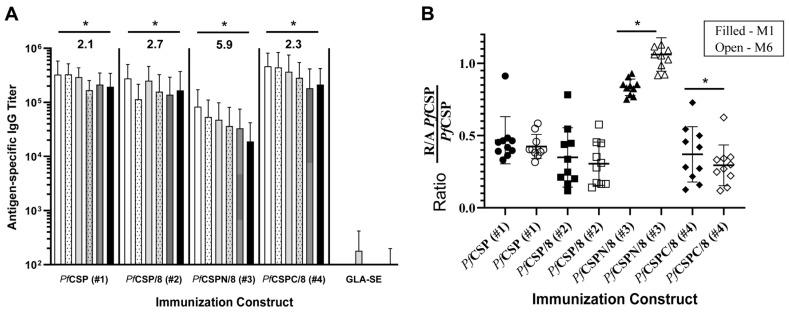
Durability of *Pf*CSP-based vaccine-induced antibodies. Groups of CD1 mice (*n* = 9–10/group, both sexes) were immunized and boosted with r*Pf*CSP (#1), r*Pf*CSP/8 (#2), r*Pf*CSPN/8 (#3), or r*Pf*CSPC/8 (#4). Following the boost, serum samples were collected once per month for a period of six months. (**A**) Sera were assessed for antigen-specific IgG responses using ELISA plates coated with r*Pf*CSP. Graph depicts mean (±SD) of antigen specific IgG titer with differences between month one (M1, white solid bars) and month six (M6, black solid bars) assessed by Wilcoxon matched pairs signed-rank test. Sequential shaded bars represent samples collected at months 2 through 5. Fold change in anti-*Pf*CSP antibody titers between M1 and M6 are indicated at the top of the figure. (**B**) Secondary sera obtained at M1 (filled symbols) and M6 (open symbols) were assayed by ELISA on plates coated with r*Pf*CSP (#1) or R/A r*Pf*CSP (#1). The proportion of antibodies in each sera recognizing linear B cell epitopes was expressed as the ratio of the titer on R/A r*Pf*CSP divided by the titer on intact r*Pf*CSP (y-axis) with statistical significance of differences in ratios determined the Wilcoxon matched pairs signed-rank test. * *p* < 0.05.

**Table 1 vaccines-12-00351-t001:** Study #4: Sterile protection induced by immunization with r*Pf*CSP-based vaccines.

Vaccine Group	# Infected/# Challenged	% Sterile	*p* Value ^a^ (Vaccine vs. Control)
r*Pf*CSP (#1)	6/10	40%	0.097
r*Pf*CSP/8 (#2)	2/9	77%	0.001
r*Pf*CSPN/8 (#3)	7/10	30%	0.393
r*Pf*CSPC/8 (#4)	6/10	40%	0.137
Control (r*Pf*MSP8 + GLA-SE)	18/20	10%	--

^a^ Mantel–Cox log rank test—time to appearance of blood-stage parasites, # of animals parasite free.

**Table 2 vaccines-12-00351-t002:** Study #5: Sterile protection induced by immunization with r*Pf*CSP-based vaccines.

Vaccine Group	# Infected/# Challenged	% Sterile	*p* Value ^a^ (Vaccine vs. Control)
r*Pf*CSP (#1)	3/10	70%	0.062
R/A r*Pf*CSP (#1)	2/10	80%	0.045
r*Pf*CSP/8 (#2)	2/10	80%	0.024
R/A r*Pf*CSP/8 (#2)	1/9	89%	0.011
Control (r*Pf*MSP8 + GLA-SE)	13/20	35%	--

^a^ Mantel–Cox log rank test—time to appearance of blood-stage parasites, # of animals parasite free.

## Data Availability

The original contributions presented in this study are included in the article/Appendix A. Further inquiries can be directed to the corresponding author.

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
