# Peer review of "Design and Evaluation of Chimeric Plasmodium falciparum Circumsporozoite Protein-Based Malaria Vaccines"

_vaccines, 2024, doi:10.3390/vaccines12040351_

Round 1

Reviewer 1 Report

Comments and Suggestions for Authors

    The work presented in the manuscript entitled “Design and evaluation of chimeric Circumsporozoite protein-based malaria vaccines” proposes to improve the efficacy and longevity of the current formulation of the malaria vaccine by adding the N-terminal domain, the central repeat region, and the C terminal domain of the P. falciparum circumsporozoite surface protein (PfCSP) and using PfMSP8 as a carrier. The constructs were tested in mice.

    Although the current malaria vaccine has been improved by use of more effective adjuvant, identifying additional conserved B cell epitopes by including the N-terminal sequence of PfCSP may increase the efficacy of the vaccine. Therefore, generating additional B cell epitopes is highly relevant to the field and the use of the P. falciparum merozoite surface protein 8, PfMSP8, as a carrier is both original and relevant to the field of malaria vaccine. Based on the data presented in the manuscript, the number of B cell epitopes was significantly higher than those recognized with the current vaccine.

    The authors have shown that the constructs containing the N-terminal domain, the central repeat region, and the C terminal domain of PfCSP-based vaccines generate high titers antibodies against PfCSP IgG of various subclasses (Figure 3). Thus, the diversity of B cell epitopes was increased. The diversity of the T cell epitopes was likewise improved (Figure 4). Importantly, the data shown in Figure 8 indicate that immunization with the chimeric rPfCSP/8 offers better protection against sporozoite challenge infection demonstrating the efficacy of the novel construct (Figure 8).
Although 10 mice were used for each construct, a larger sample size would strengthen the conclusions.

Major Concern:
The author should consider creating a version of construct #4 fused to HBsAg. That would be an appropriate control to test the effectiveness of PfMSP8 as a carrier.
The references are appropriate.

Author Response

We thank the reviewer for their positive comments and suggestions and have considered each carefully.

Consideration of a larger sample size.  We felt it was very important to use  genetically heterogeneous outbred mice for these studies to better reflect the variability in human populations.  Because of this, we expected and observed greater variability in immune responses.  Rather than increase the number of animals per group beyond n=10, we chose instead to conduct independent replicates of key, informative immunogenicity studies and assessments.  In addition and where appropriate, we combined the data from 2 or 3 biological replicates to allow analysis of responses based on 27-30 mice per construct.  This strengthened our statistical analysis and increase confidence in the conclusions drawn.  

Consider construct #4 fused to HBsAg.     We recognize the reviewer's interest in comparison of PfMSP8 vs HBsAg as a vaccine carrier for construct #4.  Our focus on the use of PfMSP8 was based on our prior published work with additional blood-stage and sexual stage vaccine candidates.  In these studies, the value of PfMSP8 as a vaccine carrier was clearly established in comprehensive comparative immunogenicity studies.  With the goal of incorporating a pre-erythrocytic stage component into our most successful trivalent vaccine formulations, we focused primarily on the design of the PfCSP targeted domain.  Our goal was not simply to improve the immunogenicity of the truncated PfCSP construct that has been tested, but to improve the diversity of epitopes recognized throughout the PfCSP.  We achieved this with our construct #2 and demonstrated a positive impact on efficacy.  We feel that comparing PfMSP8 and HBsAg is asking a different question and beyond the scope of the present effort.  Based on the data we obtained, we feel that such a comparison with construct #2 not construct #4 would be more informative and can consider this in our future studies.  We have added a line in the discussion to note the reviewer's point.

Reviewer 2 Report

Comments and Suggestions for Authors

Stump et al., present a recombinant vaccine prototype based on recombinant rPfCSP portions that are outside the ones currently used in RTS, S and R21 formation and in addition they utilised PfMSP8 protein to boost immune response. In this study author show, that N-terminal portion of PfCSP can drive a protective response and PfMSP8 can be used to obtain higher antibody titre over time. Although rPfCSP is a validated vaccine target and therefore, it is not surprising that it can elicit high titres and sustained protection, this study shows how it is important to fully characterise all the portions of an antigen to fully comprehend the induced immune response. Additionally, the usage of another P. falciparum protein resulted to be beneficial to boost the response. I do not have any reserve about the manuscript. However, I strongly recommend to increase the resolution and size of Figures. 

Author Response

We thank the reviewer for their positive comments. 

As suggested, we have increased the size of Figures 1, 2, 3, 4, 7, and 9 to the extent possible within the fixed parameters of the template.

Reviewer 3 Report

Comments and Suggestions for Authors

Stump and colleagues report on the efficacy, immunogenicity and protection obtained from the vaccine rPfCSP constructed with the carrier protein rPfMSP8. This is a well written manuscript that adds to our knowledge of the protective mechanisms of RTS,S and R21 malaria vaccines. However, response/comments regarding the questions below should be included in the discussion. The robust antibody response to the rPfMSP8 carrier compared to the rPfCSP immunogen is a concern. An explanation regarding the magnitude of the memory response generated is warranted due to the low T cell specific response obtained for rPfCSP.

1. With the low rPfCSP-specific responses to rPfCSP/8 and robust response to the carrier observed, is protection to challenge sporozoite infection biased towards the carrier?

2. How do the authors explain the levels of anti-rPfCSP/8 response for the  high number of peptides recognized (Figures 6B and 7C) in light of the robust response generated to rPfCSP/8 compared to the three other constructs investigated?

Author Response

We thank the reviewer for their positive comments and suggestions and have considered each carefully.

Low PfCSP-specific T cell response to rPfCSP/8, memory and protection.  Immunization with the rPfCSP fused to PfMSP8, allows cognate recognition of T and B cell epitopes in a single protein upon immunization. T cell recognition of strong epitopes within the PfMSP8 carrier is effective in providing help to PfCSP-specific B cells, driving strong, class-switched, durable B cell/Ab responses.  We previously demonstrated this with chimeric MSP1/8 vaccines in both P. yoelii and P. falciparum.  By inclusion of an additional group of animals immunized with the PfMSP8 carrier alone, we demonstrated that protection is dependent on the PfCSP-specific effector responses, and not the effector activity of antibodies or T cells specific for PfMSP8 (Figure 5B).  

Explanation for the high number of peptides recognized following PfCSP/8 immunization.  As noted above, the ability of PfMSP8 specific T cells to provide help to PfCSP-specific B cell combined with the use of the larger, nearly full-length PfCSP domain partly explains the increase in diversity of the B cell response.  In addition, we previously showed in studies with PfMSP2 that fusion to PfMSP8 prevents fibril formation and the masking of B cell epitopes in PfMSP2.  This increased the production of antibodies of greater diversity which were functional in the opsonophagocytosis of merozoites.  A similar influence on the conformation of PfCSP when fused to PfMSP8 may also promote an increase in the breadth of the B cell response.

We have modified to discussion to reflect consideration of these possibilities.